# Conditional Support Alignment for Domain Adaptation with Label Shift

## Abstract

Unsupervised domain adaptation (UDA) refers to a domain adaptation framework in which a learning model is trained based on the labeled samples on the source domain and unlabelled ones in the target domain. The dominant existing methods in the field that rely on the classical covariate shift assumption to learn domain-invariant feature representation have yielded suboptimal performance under label distribution shift. In this paper, we propose a novel conditional adversarial support alignment (CASA) whose aim is to minimize the conditional symmetric support divergence between the source's and target domain's feature representation distributions, aiming at a more discriminative representation for the classification task. We also introduce a novel theoretical target risk bound, which justifies the merits of aligning the supports of conditional feature distributions compared to the existing marginal support alignment approach in the UDA settings. We then provide a complete training process for learning in which the objective optimization functions are precisely based on the proposed target risk bound. Our empirical results demonstrate that CASA outperforms other state-of-the-art methods on different UDA benchmark tasks under different label shift conditions.

## 1 Introduction

The remarkable success of modern deep learning models often relies on the assumption that training and test data are independent and identically distributed (i.i.d), contrasting the types of real-world problems that can be solved. The violation of that i.i.d. assumption leads to the data distribution shift, or out-of-distribution (OOD) issue, which negatively affects the generalization performance of the learning models (Torralba & Efros, 2011; Li et al., 2017) and renders them impracticable. One of the most popular settings for the OOD problem is unsupervised domain adaptation (UDA) (Ganin & Lempitsky, 2015; David et al., 2010) in which the training process is based on fully-labeled samples from a source domain and completely-unlabeled samples from a target domain.

While the *covariate shift* assumption has been extensively studied under the UDA problem setting, with reducing the feature distribution divergence between domains as the dominant approach (Ganin & Lempitsky, 2015; Tzeng et al., 2017; Shen et al., 2018; Courty et al., 2017; Liu et al., 2019; Long et al., 2015; 2017; 2016; 2014), the *label shift* assumption (i.e., the marginal label distribution $p(y)$ varies between domains, while the conditional $p(x|y)$ is unchanged) remains vastly underexplored in comparison. Compared to the covariate shift assumption, the label shift assumption is often more reasonable in several real-world settings, e.g., the healthcare industry, where the distribution of diseases in medical diagnosis may change across hospitals, while the conditional distribution of symptoms given diseases remains unchanged.

Several UDA methods that explicitly consider the label shift assumption often rely on estimating the importance weights of the source and target label distribution and strictly require the conditional distributions $p(x|y)$ or $p(z|y)$ to be domain-invariant Lipton et al. (2018); Tachet des Combes et al. (2020); Azizzadenesheli et al. (2019). Another popular UDA under label shift framework is enforcing domain invariance of representation $z$ w.r.t some *relaxed* divergences Wu et al. (2019); Tong et al. (2022). Wu et al. (2019) proposed reducing $\beta$-admissible distribution divergence to prevent cross-label mapping in conventional domain-invariant approaches. However, choosing inappropriate $\beta$ values can critically reduce the performance of this method under extreme label shifts (Wu et al., 2019; Li et al., 2020).

This paper aims to develop a new theoretically sound, namely conditional adversarial support alignment (CASA) approach for UDA under the label shift. Our proposed method is relatively related to but different from the adversarial support alignment (ASA) (Tong et al., 2022) one, which utilizes the symmetric support divergence (SSD) to align the support of the marginal feature distribution of the source and the target domains. One of the critical drawbacks of the ASA method is that reducing the marginal support divergence indiscriminately may make the learned representation susceptible to conditional distribution misalignment. Our proposed CASA alleviates that issue by considering discriminative features when aligning the supports of two distributions. In particular, the conditional support alignment instead of marginal case makes CASA less susceptible to misalignment of features between different classes than ASA, which is illustrated intuitively by Figure 1.

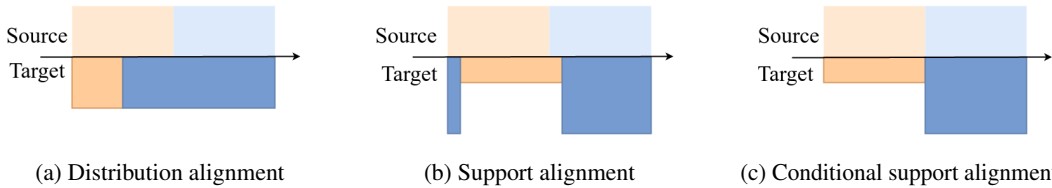

(a) Distribution alignment      (b) Support alignment      (c) Conditional support alignment

Figure 1: Illustration of the learned latent space of different domain-invariant frameworks under label shift for a binary classification problem. It can be seen that the support alignment (b) can mitigate the high error rate induced by distribution alignment (a), whereas the conditional support alignment (c) can achieve the best representation by explicitly aligning the supports of class-conditioned latent distributions.

The main contributions of our paper are summarized as follows:

- We propose a novel conditional adversarial support alignment (CASA) to align the support of the conditional feature distributions on the source and target domains, aiming for a more label-informative representation for the classification task.

- We provide a new theoretical upper bound for the target risk for the learning process of our CASA. We then introduce a complete training scheme for our proposed CASA by minimizing that bound.

- We provide experimental results on several benchmark tasks in UDA, which consistently demonstrate the empirical benefits of our proposed method compared to other relevant existing UDA approaches.

## 2 METHODOLOGY

### 2.1 PROBLEM STATEMENT

Let us consider a classification framework where $\mathcal{X} \subset \mathbb{R}^d$ represents the input space and $\mathcal{Y} = \{y_1, y_2, \ldots, y_K\}$ denotes the output space. A domain is then defined by $P(x, y) \in \mathcal{P}_{\mathcal{X} \times \mathcal{Y}}$, where $\mathcal{P}_{\mathcal{X} \times \mathcal{Y}}$ is the set of joint probability distributions on $\mathcal{X} \times \mathcal{Y}$. The set of conditional distributions of $x$ given $y$, $P(x|y)$, is denoted as $\mathcal{P}_{\mathcal{X}|\mathcal{Y}}$, and the set of probability marginal distributions on $\mathcal{X}$ and $\mathcal{Y}$ is denoted as $\mathcal{P}_{\mathcal{X}}$ and $\mathcal{P}_{\mathcal{Y}}$, respectively. We also denote $P_{X|Y=y_k}$ as $P_{X|k}$ for convenience.

Consider an UDA framework, in which the (labelled) source domain $D^S = \left\{ \left( x_i^S, y_i^S \right) \right\}_{i=1}^{n_S}$, where $\left( x_i^S, y_i^S \right) \sim P^S(x, y)$, and the (unlabelled) target domain $D^T = \left\{ x_j^T \right\}_{j=1}^{n_T}$, where $x_j^t \sim P_X^T \subset \mathcal{P}_{\mathcal{X}}$. Without loss of generality, we assume that both the source and target domains consists of $K$ classes, i.e., $\mathcal{Y} = \{y_1, \ldots, y_K\}$. In this paper, we focus on the UDA setting with label shift, which assumes that $P_Y^S \neq P_Y^T$ while the conditional distributions $P_{X|Y}^S$ and $P_{X|Y}^T$ remain unchanged. Nevertheless, unlike relevant works such as Tachet des Combes et al. (2020); Lipton et al. (2018), we do not make the strong assumption about the invariance of $P_{X|Y}$ or $P_{Z|Y}$ between those domains, targeting more general UDA settings under label shift.

A classifier (or hypothesis) is defined by a function $h : \mathcal{X} \mapsto \Delta_K$, where $\Delta_K = \left\{ \boldsymbol{\pi} \in \mathbb{R}^K : \|\boldsymbol{\pi}\|_1 = 1 \wedge \pi \geq \mathbf{0} \right\}$ is the $K$-simplex, and an induced scoring function $g : \mathcal{X} \mapsto \mathbb{R}^K$.

Consider a loss function $\ell : \mathbb{R}^K \times \mathbb{R}^K \to \mathbb{R}_+$, satisfying $\ell(y, y) = 0, \ \forall y \in \mathcal{Y}$. Given a scoring function $g$, we define its associated classifier as $h_g$, i.e., $h_g(\mathbf{x}) = \hat{y}$ with $\hat{y} \in \arg\min_{y \in \mathcal{Y}} \ell[g(\mathbf{x}), y]$. For conciseness, we consider any hypothesis $h$ also a scoring function $g$.

The $\ell$-risk of a scoring function $g$ over a distribution $P_{X \times Y}$ is then defined by $\mathcal{L}_P(g) := \mathbb{E}_{(x,y) \sim P}[\ell(g(x), y)]$, and the classification mismatch of $g$ with a classifier $h$ by $\mathcal{L}_P(g, h) := \mathbb{E}_{x \sim P_X}[\ell(g(x), h(x))]$. For convenience, we denote the source and target risk of scoring function or classifier $g$ as $\mathcal{L}_S(g)$ and $\mathcal{L}_T(g)$, respectively.

## 2.2 A TARGET RISK BOUND BASED ON SUPPORT MISALIGNMENT

Similar to Ganin et al. (2016), we assume that the hypothesis $g$ can be decomposed as $g = c \circ f$, where $c : \mathcal{Z} \to \mathcal{Y}$ is the classifier, $f : \mathcal{X} \to \mathcal{Z}$ is the feature extractor, and $\mathcal{Z}$ represents the latent space. Let us denote the domain discriminator as $\phi : \mathcal{Z} \to \{0, 1\}$, and the marginal distribution of $Z \in \mathcal{Z}$ in source and target domains as $P_Z^S$ and $P_Z^T$, respectively.

We next introduce several necessary definitions and notations, for the theoretical development of the target error bound in our proposed CASA. Some of them are employed in the construction of the IMD-based domain adaptation bound in Dhouib & Maghsudi (2022).

**Definition 1** (*Source-guided uncertainty* (Dhouib & Maghsudi, 2022)). Let $\mathbb{H}$ be a hypothesis space, and let $\ell$ be a given loss function. The source-guided uncertainty of $g \in \mathbb{H}$ associated with $\ell$ is defined by:

$$\mathcal{C}_{\mathbb{H}}(g) = \inf_{h \in \mathbb{H}} \mathcal{L}_T(g, h) + \mathcal{L}_S(h), \tag{1}$$

where $\mathcal{L}_T(g, h)$ is the classification mismatch of $g$ and $h$ on $P_X^T$.

*Remark* 1. When $\ell$ is the cross entropy loss, minimizing the conditional entropy of predictor on the target domain, along with $\mathcal{L}_S(h_g)$, effectively minimizes $\mathcal{C}_{\mathbb{H}}(g)$ (Dhouib & Maghsudi, 2022).

**Definition 2** (*Integral measure discrepancy*). Let $\mathbb{F}$ be a family of nonnegative functions over $\mathbb{X}$, containing the null function, e.g., $\mathbb{F} = \{\ell(h, f_s); h \in \mathbb{H}\}$ with $f_s$ as the source labeling function. The Integral Measure Discrepancy (IMD) associated to $\mathbb{F}$ between two distribution $\mathcal{Q}$ and $\mathcal{P}$ over $\mathbb{X}$ is

$$\mathrm{IMD}_{\mathbb{F}}(\mathcal{Q}, \mathcal{P}) := \sup_{f \in \mathbb{F}} \int f \, \mathrm{d}\mathcal{Q} - \int f \, \mathrm{d}\mathcal{P}. \tag{2}$$

Intuitively, this discrepancy aims to capture the distances between measures w.r.t. difference masses.

**Definition 3** (*Symmetric support divergence*). Assuming that $d$ is a proper distance on the latent space $\mathcal{Z}$. The symmetric support divergence (SSD) between two probability distributions $P_Z$ and $Q_Z$ is then defined by:

$$\mathcal{D}_{\mathrm{supp}}(P_Z, Q_Z) = \mathbb{E}_{z \sim P_Z}[d(z, \mathrm{supp}(Q_Z))] + \mathbb{E}_{z \sim Q_Z}[d(z, \mathrm{supp}(P_Z))],$$

where $\mathrm{supp}(P_Z)$ and $\mathrm{supp}(Q_Z)$ are the corresponding supports of $P_Z$ and $Q_Z$, respectively.

Different from the work in Dhouib & Maghsudi (2022) that extends the bound with $\beta$-admissible distances, our proposed method goes beyond interpreting the bound solely in terms of support divergences Tong et al. (2022). In particular, our method also incorporates the label structures of the domains, allowing a more comprehensive analysis of the underlying relationships.

## 2.3 A NOVEL CONDITIONAL SSD-BASED DOMAIN ADAPTATION BOUND

In this work, we first introduce a definition for a novel *conditional symmetric support divergence* between the conditional distributions $P_{Z|Y}^S$ and $P_{Z|Y}^T$. For simplicity, we also denote $d$ as a well-defined distance on the conditional $\mathcal{Z}|\mathcal{Y}$ space.

**Definition 4** (*Conditional symmetric support divergence*). The conditional symmetric support divergence (CSSD) between the conditional distributions $P_{Z|Y}^S$ and $P_{Z|Y}^T$ is defined by

$$\mathcal{D}_{\mathrm{supp}}^c(P_{Z|Y}^S, P_{Z|Y}^T)$$
$$= \sum_{y \in \mathcal{Y}} P^S(Y = y) \mathbb{E}_{z \sim P_{Z|Y=y}^S}[d(z, \mathrm{supp}(P_{Z|Y=y}^T))] + P^T(Y = y) \mathbb{E}_{z \sim P_{Z|Y=y}^T}[d(z, \mathrm{supp}(P_{Z|Y=y}^S))].$$

We establish the justification for CSSD as a support divergence through the following result.

**Proposition 1.** *Assume $P^S(Y = y) > 0, P^T(Y = y) > 0$ for any $y \in \mathcal{Y}$, $\mathcal{D}^c_{\mathrm{supp}}(P^S_{Z|Y}, P^T_{Z|Y})$ is a support divergence.*

The proof is deferred to the Appendix. In comparison to the SSD in Definition 3, our CSSD takes into account the class proportions in both source and target domains. As a result, the localized IMD considers per-class localized functions, which are defined as the $(\epsilon, P^S_{Z|Y})$-localized nonnegative function denoted by $\mathbb{F}_\epsilon$. Specifically, $\mathbb{F}_\epsilon = \{f; f(z) \geq 0, \mathbb{E}_{P^S_{Z|k}}[f] \leq \epsilon_k, \quad k = 1 \ldots K\}$ with $\epsilon = (\epsilon_1, \ldots, \epsilon_K) \geq 0$. In the following lemma, we introduce upper bounds for the IMD using CSSD (the corresponding proof is provided in the Appendix A).

**Lemma 1** (Upper bound IMD using CSSD). *Let $\mathbb{F}$ be a set of nonnegative and $1$-Lipschitz functions, and let $\mathbb{F}_\epsilon$ be a $(\epsilon, P^S_{Z|Y})$-localized nonnegative function. Then, we can bound the IMD w.r.t the conditional support domain and CSSD, respectively, as*

$$\mathrm{IMD}_{\mathbb{F}_\epsilon}(P^T_Z, P^S_Z) \leq \sum_{k=1}^K q_k \mathbb{E}_{z \sim P^T_{Z|k}}[d(z, \mathrm{supp}(P^S_{Z|k}))] + q_k \delta_k + p_k \epsilon_k, \tag{3}$$

$$\mathrm{IMD}_{\mathbb{F}_\epsilon}(P^T_Z, P^S_Z) \leq \mathcal{D}^c_{\mathrm{supp}}(P^T_{Z|Y}, P^S_{Z|Y}) + \sum_{k=1}^K q_k \delta_k + p_k \gamma_k, \tag{4}$$

*where $\delta_k := \sup_{z \in \mathrm{supp} P^S_{Z|k}, f \in \mathbb{F}_\epsilon} f(z)$, $\gamma_k := \sup_{z \in \mathrm{supp} P^T_{Z|k}, f \in \mathbb{F}_\epsilon} f(z)$, $p_k = P^S(Y = y_k)$ and $q_k = P^T(Y = y_k)$.*

We now provide a novel upper bound of the target risk based on CSSD in the following theorem, which is a straightforward result from Lemma 1.

**Theorem 1** (Domain adaptation bound via CSSD). *Let $\mathbb{H}$ be a hypothesis space, $g$ be a score function, and $\ell$ be a loss function satisfying the triangle inequality. Consider the localized hypothesis $\mathbb{H}^\mathbf{r} := \{h \in \mathbb{H}; \mathcal{L}_{S_k}(h) \leq r_k, k = 1 \ldots K\}$. Given that all the assumptions for $\mathbb{F}_\epsilon$ in Lemma 1 are fulfilled. Then, for any $\mathbf{r}^1 = (r^1_1, \ldots, r^1_K) \geq 0, \mathbf{r}^2 = (r^2_1, \ldots, r^2_K) \geq 0$ that satisfy $r^1_k + r^1_k = \epsilon_k$, we have:*

$$\mathcal{L}_T(g) \leq \mathcal{C}_{\mathbb{H}^{\mathbf{r}^1}}(g) + \mathcal{D}^c_{\mathrm{supp}}(P^T_{Z|Y}, P^S_{Z|Y}) + \sum_{k=1}^K q_k \delta_k + p_k \gamma_k + \inf_{h \in \mathbb{H}^{\mathbf{r}^2}} \mathcal{L}_S(h) + \mathcal{L}_T(h). \tag{5}$$

*Remark* 2. In the case that we do not incorporate the label information, the IMD can be bounded as

$$\mathrm{IMD}_{\mathbb{F}_\epsilon} \leq \mathbb{E}_{z \sim P^T_Z}[d(z, \mathrm{supp}(P^S_Z))] + \delta + \epsilon, \tag{6}$$

where $\delta = \sup_{z \in \mathrm{supp}(P^S_Z), f \in \mathbb{F}_\epsilon} f(z)$. Similar to the findings in Dhouib & Maghsudi (2022), the inequality in Equation 6 provides a justification for minimizing SSD proposed in Tong et al. (2022). Notably, this inequality extends to the case where $\epsilon \geq 0$, and thus recover the bound with SSD in Dhouib & Maghsudi (2022) as a special case. Note that in order to make a fair comparison between Equation 3 and 6, we assume that $\sum_k \epsilon_k p_k = \epsilon$ making $\mathbb{F}_\epsilon \subseteq \mathbb{F}_\epsilon$.

In comparison the the upper bound in Equation 6, our expectation

$$\sum_k q_k \mathbb{E}_{z \sim P^T_{Z|k}}[d(z, \mathrm{supp}(P^S_{Z|k}))] \geq \mathbb{E}_{z \sim P^T_Z}[d(z, \mathrm{supp}(P^S_Z))],$$

due to the fact that $d(z, \mathrm{supp}(P^S_{Z|k})) \geq d(z, \mathrm{supp}(P^S_Z))$ and that Jensen's inequality holds.

However, this inequality does not imply that our bound is less tight than the bound using SSD. When considering the remaining part, we can observe that $\sum_k q_k \delta_k \leq \delta$ since $\mathrm{supp}(P^S_{Z|k}) \subseteq \mathrm{supp}(P^S_Z)$ for any $k = 1, \ldots, K$. In other words, there is a trade-off between the distance to support space and the uniform norm (sup norm) of function on the supports.

*Remark* 3. The proposed bound shares several similarities with other target error bounds in the UDA literature (Ben-David et al., 2010; Acuna et al., 2021). In particular, these bounds all upperbound the

target risk with a source risk term, a domain divergence term, and an ideal joint risk term. The main difference is that we use the conditional symmetric support divergence instead of $\mathcal{H}\Delta\mathcal{H}$-divergence in Ben-David et al. (2010) and $f$-divergence in Acuna et al. (2021), making our bound more suitable for problems with large degrees of label shift, as a lower value of CSSD does not necessarily increase the target risk under large label shift, unlike $\mathcal{H}\Delta\mathcal{H}$-divergence and $f$-divergence (Tachet des Combes et al., 2020; Zhao et al., 2019). Furthermore, the localized hypothesis spaces $\mathbb{H}^{\mathbf{r}^1}$ and $\mathbb{H}^{\mathbf{r}^2}$ are reminiscent of the localized adaptation bound proposed in Zhang et al. (2020). While lower values of $\mathbf{r}^1, \mathbf{r}^2$ can make the term $\sum_{k=1}^{K} q_k \delta_k + p_k \gamma_k$ smaller, the source-guided uncertainty term and ideal joint risk term can increase as a result. In our final optimization procedure, we assume the ideal joint risk term and $\sum_{k=1}^{K} q_k \delta_k + p_k \gamma_k$ values to be small and minimize the source-guided uncertainty (see section 2.4.1) and CSSD via another proxy (see section 2.4.3) to reduce the target domain risk.

## 2.4 TRAINING SCHEME FOR OUR CASA

So far, we have presented the main ideas of our CASA algorithm in a general manner. In the next section, we discuss the implementation details of our proposed framework.

### 2.4.1 MINIMIZING SOURCE-GUIDED UNCERTAINTY

As stated in Remark 1, minimizing the source risk and the target conditional entropy reduces the source-guided uncertainty $\mathcal{C}_{\mathbb{H}^{\mathbf{r}_1}}(g)$, which is the second term in the target risk bound of equation 5. Minimizing the prediction entropy has also been extensively studied and resulted in effective UDA algorithms (Shu et al., 2018; Kirchmeyer et al., 2022; Liang et al., 2021). Hence, the total loss of CASA first includes the overall classification loss $\mathcal{L}_y(\cdot)$ on source samples and the conditional entropy loss on target samples $\mathcal{L}_{ce}(\cdot)$, defined as follows:

$$\mathcal{L}_y(g) = \frac{1}{n_S} \sum_{i=1}^{n_S} \ell(g(x_i^S), y_i^S) \quad \text{and} \quad \mathcal{L}_{ce}(g) = -\frac{1}{n_T} \sum_{i=1}^{n_T} g(x_i^T)^\top \ln g(x_i^T). \tag{7}$$

### 2.4.2 ENFORCING LIPSCHITZ HYPOTHESIS

The risk bound in Eq. 5 suggests regularizing the Lipschitz continuity of the classifier $c$. Inspired by the success of virtual adversarial training by Miyato et al. (2018) on domain adaptation tasks (Shu et al., 2018; Tong et al., 2022), we instead enforce the locally-Lipschitz constraint of the classifier, which is a relaxation of the global Lipschitz constraint, by enforcing consistency in the norm-ball w.r.t each representation sample $z$. In additional, we observe that enforcing the local Lipschitz constraint of $g = c \circ f$ instead of $c$ leads to better performance in empirical experiments. Hence, we introduce the virtual adversarial loss term (Miyato et al., 2018), which enforces the classifier consistency within the $\epsilon$-radius neighborhood of each sample $x$ by penalizing the KL-divergence between predictions of nearby samples, and follow the approximation method in Miyato et al. (2018)

$$\mathcal{L}_v(c, f) = \frac{1}{n_S} \sum_{i=1}^{n_S} \max_{\|r\| < \epsilon} D_{\mathrm{KL}}\left(g(x_i^S) \| g(x_i^S + r)\right) + \frac{1}{n_T} \sum_{i=1}^{n_T} \max_{\|r\| < \epsilon} D_{\mathrm{KL}}\left(g(x_i^T) \| g(x_i^T + r)\right).$$
$$\tag{8}$$

### 2.4.3 MINIMIZING CONDITIONAL SYMMETRIC SUPPORT DIVERGENCE

The next natural step for reducing the target risk bound in equation 5 is to minimize $\mathcal{D}_{\mathrm{supp}}^c(P_{Z|Y}^S, P_{Z|Y}^T)$. However, it is challenging to directly optimize this term since in a UDA setting, we have no access to the labels of the target samples. Motivated by the use of pseudo-labels to guide the training process in domain adaptation literature (French et al., 2018; Chen et al., 2019; Long et al., 2018; Zhang et al., 2019a), we alternatively consider minimizing $\mathcal{D}_{\mathrm{supp}}^c(P_{Z|\widehat{Y}}^S, P_{Z|\widehat{Y}}^T)$ as a proxy for minimizing $\mathcal{D}_{\mathrm{supp}}^c(P_{Z|Y}^S, P_{Z|Y}^T)$, where $\widehat{Y}$ are pseudo-labels. To mitigate the error accumulation issue of using pseudolabels under large domain shift (Zhang et al., 2019a; Liu et al., 2021), we employ the entropy conditioning technique in Long et al. (2018) in our implementation of CASA . Nevertheless, given that the estimation of the conditional support alignment $\mathcal{D}_{\mathrm{supp}}^c(P_{Z|Y}^S, P_{Z|Y}^T)$ is based on the approximation of $P_Y^T$, which is commonly time-consuming and error-prone, the following

---

**Algorithm 1** Conditional Adversarial Support Alignment

---

**Input**: $D^S = \left\{ \left( x_i^S, y_i^S \right) \right\}_{i=1}^{n_S}, D^T = \left\{ x_j^T \right\}_{j=1}^{n_T}$

**Output**: Feature extractor $f$, classifier $c$, domain discriminator $r$

1: **for** number of training iterations **do**
2:     Sample minibatch from source $\left\{ \left( x_i^S, y_i^S \right) \right\}_{i=1}^{m}$ and target $\left\{ x_i^T \right\}_{i=1}^{m}$
3:     Update $\phi$ according to Eq. 12
4:     Update $f, c$ according to Eq. 11
5: **end for**

---

proposition motivates us to alternatively minimize the joint support divergence $\mathcal{D}_{\text{supp}}(P_{Z,Y}^S, P_{Z,Y}^T)$ to tighten the target error bound, without using any explicit estimate of the marginal label distribution shift as performed in Lipton et al. (2018); Tachet des Combes et al. (2020).

**Proposition 2.** *Assuming that $P^S(\widehat{Y} = y) > 0$, $P^T(\widehat{Y} = y) > 0, \forall y \in \mathcal{Y}$, and there exists a well-defined distance denoted by $d$ in the space $\mathcal{Z} \times \mathcal{Y}$. Then $\mathcal{D}_{\text{supp}}^c(P_{Z|Y}^S, P_{Z|Y}^T) = 0$ if and only if $\mathcal{D}_{\text{supp}}(P_{Z,Y}^S, P_{Z,Y}^T) = 0$.*

Moreover, minimizing this joint support divergence can be performed efficiently in one-dimensional space. In particular, Tong et al. (2022) indicated that when considering the log-loss discriminator $\phi : \mathcal{X} \to [0, 1]$ trained to discriminate between two distributions $P$ and $Q$ with binary cross entropy loss function can be can be used to estimate $\mathcal{D}_{\text{supp}}(P, Q)$. Instead of aligning the marginal distributions $P_Z^S$ and $P_Z^T$, our approach focuses on matching the support of joint distributions, $P_{Z,\widehat{Y}}^S$ and $P_{Z,\widehat{Y}}^T$. We use a trained optimal discriminator $\phi^*$ to discriminate between these distributions, which are represented as the outer product $Z \otimes \widehat{Y}$ (Long et al., 2018). Consequently, our model incorporates the domain discriminator loss and support alignment loss to minimize the conditional support divergence $\mathcal{D}_{\text{supp}}^c(P_{Z|Y}^S, P_{Z|Y}^T)$ in the error bound specified in equation 5.

$$\mathcal{L}_d(\phi) = -\frac{1}{n_S} \sum_{i=1}^{n_S} \ln \left[ G(x_i^S) \right] - \frac{1}{n_T} \sum_{i=1}^{n_T} \ln \left[ 1 - G(x_i^T) \right]; \tag{9}$$

$$\mathcal{L}_{align}(f) = \frac{1}{n_S} \sum_{i=1}^{n_S} d \left( G(x_i^S), \{ G(x_j^T) \}_{j=1}^{n_T} \right) + \frac{1}{n_T} \sum_{i=1}^{n_T} d \left( G(x_i^T), \{ G(x_j^S) \}_{j=1}^{n_S} \right), \tag{10}$$

where $G(x) = \phi(f(x) \otimes g(x))$ and $u \otimes v = uv^T$. Here, similar to Tong et al. (2022), $d(\cdot, \cdot)$ is either the squared L2 or the L1 distance.

Overall, the training process of our proposed algorithm, CASA , can be formulated as an alternating optimization problem (see Algorithm 1),

$$\min_{f,c} \mathcal{L}_y(g) + \lambda_{align} \mathcal{L}_{align}(f) + \lambda_{ce} \mathcal{L}_{ce}(g) + \lambda_v \mathcal{L}_v(g), \tag{11}$$

$$\min_{\phi} \mathcal{L}_d(\phi), \tag{12}$$

where $\lambda_{align}, \lambda_y, \lambda_{ce}, \lambda_v$ are the weight hyper-parameters associated with the respective loss terms.

## 3 EXPERIMENTS

### 3.1 SETUP

**Datasets.** We focus on visual domain adaptation tasks and empirically evaluate our proposed algorithm CASA on benchmark UDA datasets **USPS → MNIST**, **STL → CIFAR** and **VisDA-2017**. We further conduct experiments on the **DomainNet** dataset and provide the results in Appendix due to the page limitation. For VisDA-2017 and DomainNet, instead of using the 100% unlabeled target data for both training and evaluation, we utilize 85% of the unlabeled target data for training, and the remaining 15% for evaluation, to mitigate potential overfitting (Garg et al., 2023).

**Evaluation setting.** To assess CASA 's robustness to label shift, we adopt the experimental protocol of Garg et al. (2023). We simulate label shift using the Dirichlet distribution, keeping the source

label distribution unchanged and $P_Y^T(y) \sim Dir(\beta)$, with $\beta_y = \alpha.P_Y^{T0}(y)$, $P_Y^{T0}(y)$ as the original target marginal, and $\alpha$ values of $10, 3.0, 1.0, 0.5$. We also include a no label shift setting, denoted as $\alpha = $ None, where both source and target label distributions are unchanged. For each method and label shift degree, we perform 5 runs with different random seeds and report average per-class accuracy on the target domain's test set as evaluation metrics.

**Baselines.** We assess the performance of CASA by comparing it with various existing UDA algorithms, including: No DA (training using solely labeled source samples), DANN (Ganin et al., 2016), CDAN (Long et al., 2018), VADA (Shu et al., 2018), SDAT (Rangwani et al., 2022), MIC (Hoyer et al., 2023), IWDAN, IWCDAN (Tachet des Combes et al., 2020), sDANN (Wu et al., 2019), ASA (Tong et al., 2022), PCT (Tanwisuth et al., 2021), and SENTRY (Prabhu et al., 2021). Whereas IWDAN and IWCDAN rely on importance weighting methods, CDAN, IWCDAN, FixMatch and SENTRY employ target pseudolabels. Moreover, we apply the resampling and reweighting heuristics in Garg et al. (2023) to DANN, CDAN and FixMatch to make these baselines more robust to label shift, and denote them as DANN*, CDAN* and FixMatch* in Table 1, 2 and 3. Further implementation details, including the hyperpamarater values and network architectures, are provided in the Appendix.

## 3.2 MAIN RESULTS

We report the results on USPS→MNIST, STL→CIFAR and VisDA-2017 in Table 1, 2 and 3 respectively.

Table 1: Per-class average accuracies on USPS→MNIST. Bold and underscore denote the best and second-best methods respectively.

| Algorithm | $\alpha = $ None | $\alpha = 10$ | $\alpha = 3.0$ | $\alpha = 1.0$ | $\alpha = 0.5$ | Average |
|---|---|---|---|---|---|---|
| No DA | 73.9 | 73.8 | 73.5 | 73.9 | 73.8 | 73.8 |
| DANN* | 96.2 | 96.2 | 93.5 | 82.6 | 72.3 | 88.2 |
| CDAN* | 96.6 | 96.5 | 93.7 | 82.2 | 70.7 | 88.0 |
| VADA | **98.1** | **98.1** | 96.8 | 84.9 | 76.8 | 90.9 |
| SDAT | 97.8 | 97.5 | 94.1 | 83.3 | 68.8 | 88.3 |
| MIC | 98.0 | 97.6 | 95.2 | 82.9 | 70.8 | 88.9 |
| IWDAN | 97.5 | 97.1 | 90.4 | 81.3 | 73.3 | 87.9 |
| IWCDAN | 97.8 | 97.5 | 91.4 | 82.6 | 73.8 | 88.7 |
| sDANN | 87.4 | 90.7 | 92.1 | 89.4 | 85.2 | 89.0 |
| ASA | 94.1 | 93.7 | 94.1 | 90.8 | 84.7 | 91.5 |
| PCT | 97.4 | 97.2 | 94.3 | 82.3 | 71.8 | 88.6 |
| SENTRY | 97.5 | 91.5 | 91.4 | 84.7 | 82.3 | 89.5 |
| CASA (Ours) | 98.0 | **98.0** | **97.2** | **96.7** | **88.3** | **95.6** |

Table 2: Per-class accuracy on STL→CIFAR. Same setup as Table 1.

| Algorithm | $\alpha = $ None | $\alpha = 10$ | $\alpha = 3.0$ | $\alpha = 1.0$ | $\alpha = 0.5$ | Average |
|---|---|---|---|---|---|---|
| No DA | 69.9 | 69.8 | 69.7 | 68.8 | 67.9 | 69.2 |
| DANN* | 75.9 | 74.9 | 74.4 | 72.7 | 70.5 | 73.7 |
| CDAN* | 75.6 | 74.3 | 74.0 | 72.8 | 70.7 | 73.5 |
| VADA | **77.1** | 75.5 | 73.8 | 71.3 | 68.0 | 73.1 |
| SDAT | 75.8 | 74.4 | 71.5 | 68.3 | 66.2 | 71.2 |
| MIC | 76.4 | 75.5 | 72.7 | 68.6 | 67.3 | 72.1 |
| IWDAN | 72.9 | 72.6 | 71.8 | 70.6 | 69.5 | 71.5 |
| IWCDAN | 72.1 | 72.0 | 71.5 | 71.9 | 69.9 | 71.5 |
| sDANN | 72.8 | 72.0 | 72.0 | 71.4 | 70.1 | 71.7 |
| ASA | 72.7 | 72.2 | 72.1 | 71.5 | 69.8 | 71.7 |
| PCT | 75.0 | 76.1 | 75.0 | 70.9 | 68.3 | 73.1 |
| SENTRY | 76.7 | 76.6 | 75.2 | 71.2 | 67.0 | 73.3 |
| CASA (Ours) | 76.9 | **76.8** | **75.8** | **74.2** | **71.7** | **75.1** |

Among the methods focusing on distribution alignment, such as DANN, CDAN, and VADA, they tend to achieve the highest accuracy scores under $\alpha = $ None. However, their performances degrade significantly as the serverity of label shift increases. For instance, under $\alpha = 0.5$ in USPS→MNIST task, SDAT and MIC perform worse than source-only training by 5.0% and 3.0%, respectively.

On the other hand, baseline methods in the third group that explicitly address label distribution shift, such as ASA, sDANN and IWCDAN, often outperform distribution alignment methods under severe label shift ($\alpha \in \{1.0, 0.5\}$). However, they fall behind previous domain-invariant methods when label shift is mild or nonexistent ($\alpha \in \{$None, $10.0\}$) by a large margin of 2-4% in the STL→CIFAR task. In contrast, CASA outperforms baseline methods on 11 out of 15 transfer tasks. It achieves the second-highest average accuracies when there is no label shift in the USPS→MNIST and STL→CIFAR tasks, and outperforms the second-best methods by 3.6%, 1.6% and 0.7% under $\alpha = 0.5$ in the USPS→MNIST, STL→CIFAR and VisDA-2017 tasks, respectively.

## 3.3 VISUALIZATION AND HYPERPARAMETER ANALYSIS

**Analysis of individual loss terms.** To study the impact of each loss term in Eq.equation 11, we provide additional experiment results, which consist of the average accuracy over 5 different random

Table 3: Per-class accuracies on VisDA-2017. Same setup as Table 1.

Table 4: Ablation study of individual loss terms.

| Algorithm | $\alpha = $ None | $\alpha = 10$ | $\alpha = 3.0$ | $\alpha = 1.0$ | $\alpha = 0.5$ | Average |
|---|---|---|---|---|---|---|
| No DA | 55.6 | 56.0 | 55.5 | 55.2 | 55.1 | 55.5 |
| DANN* | 75.5 | 71.3 | 68.4 | 62.2 | 56.4 | 66.8 |
| CDAN* | 75.0 | 72.5 | 69.8 | 61.3 | 56.3 | 67.0 |
| FixMatch* | 71.6 | 67.5 | 65.6 | 60.1 | 58.7 | 64.7 |
| VADA | 75.2 | 72.3 | 69.6 | 59.2 | 52.6 | 65.8 |
| SDAT | 75.4 | 73.3 | 66.8 | 63.9 | 61.8 | 68.3 |
| MIC | **75.6** | **74.5** | 69.5 | 64.8 | 62.0 | 69.3 |
| IWDAN | 74.1 | 73.3 | 71.4 | 65.3 | 59.7 | 68.8 |
| IWCDAN | 73.5 | 72.5 | 69.6 | 62.9 | 57.2 | 67.1 |
| sDANN | 72.8 | 72.2 | 71.2 | 64.9 | 62.5 | 68.7 |
| ASA | 66.4 | 65.3 | 64.6 | 61.7 | 60.1 | 63.6 |
| PCT | 68.2 | 66.8 | 65.4 | 60.5 | 53.6 | 63.3 |
| SENTRY | 67.5 | 64.5 | 57.6 | 53.4 | 52.6 | 59.1 |
| CASA (Ours) | 74.3 | 73.4 | **71.8** | **66.3** | **63.2** | **69.8** |

| $\mathcal{L}_{align}$ | $\mathcal{L}_{ce}$ | $\mathcal{L}_v$ | $\alpha = $ None | $\alpha = 10$ | $\alpha = 3.0$ | $\alpha = 1.0$ | $\alpha = 0.5$ | Average |
|---|---|---|---|---|---|---|---|---|
| ✓ | | | 94.5 | 94.2 | 94.3 | 94.3 | 84.9 | 92.4 |
| ✓ | ✓ | | 97.7 | 97.2 | 96.8 | 96.2 | 87.4 | 95.1 |
| ✓ | ✓ | ✓ | **98.0** | **98.0** | **97.2** | **96.7** | **88.3** | **95.6** |

runs on USPS→MNIST in Table 4. It is evident that the conditional support alignment loss term $\mathcal{L}_{align}$, conditional entropy loss term $\mathcal{L}_{ce}$ and virtual adversarial loss term $\mathcal{L}_v$ all improve the model's performance across different levels of label shift.

**Visualization of learned feature embeddings under severe label shift.** We first conduct an experiment to visualize the effectiveness of our proposed method. Samples from three classes (3, 5, and 9) are selected from USPS and MNIST datasets, following Tong et al. (2022), to create source and target domains, respectively. The label probabilities are equal in the source domain, while they are $[22.9\%, 64.7\%, 12.4\%]$ in the target domain. We compare the per-class accuracy scores, Wasserstein distance $\mathcal{D}_W$, $\mathcal{D}_{\text{supp}}^c$ and 2D feature distribution of CDAN, ASA and CASA.

Fig. 3 shows that CASA achieves a higher target average accuracy, resulting in a clearer separation among classes and more distinct feature clusters compared to CDAN and ASA. Although ASA reduces the overlap of different target classes to some extent, its approach that only enforces support alignment between marginals does not fully eradicate the overlap issue. CASA tackles this drawback by considering discriminative class information during the support alignment in feature embeddings. The plot also demonstrates that CASA effectively reduces $\mathcal{D}_{\text{supp}}^c$ through the proxy objective in Proposition 2. Our observations are consistent with those made in Tong et al. (2022), namely that lower values of $\mathcal{D}_{\text{supp}}^c$ tend to correlate with higher accuracy values under label distribution shift. This visualization helps explain the superior performance of CASA over other baseline methods under severe label shift settings.

**Hyperparameter analysis** We analyze the impact of hyperparameters $\lambda_{align}$, $\lambda_{ce}$ and $\lambda_v$ on the performance of CASA on the task USPS→MNIST, with $\alpha = 1.0$. Overall, the performance remains stable as $\lambda_{align}$ increases, reaching a peak at $\lambda_{align} = 1.5$. On the other hand, the model's accuracy increases sharply at lower values of $\lambda_{ce}$, $\lambda v$ and plunges at values greater than 0.1. This means that choosing appropriate values of these 2 hyperparameters may require more careful tuning compared to $\lambda_{align}$.

## 4 RELATED WORKS

A dominant approach for tackling the UDA problem is learning domain-invariant feature representation, based on the theory of Ben-David et al. (2006), which suggests minimizing the $\mathcal{H}\Delta\mathcal{H}$-divergence between the two marginal distributions $P_Z^S$, $P_Z^T$. More general target risk bound than that of Ben-David et al. (2006) have been developed by extending the problem setting to multi-source domain adaptation (Mansour et al., 2008; Zhao et al., 2018; Phung et al., 2021), or considering discrepancy distance (Mansour et al., 2009), hypothesis-independent disparity discrepancy (Zhang et al., 2019b), or PAC-Bayesian bounds with weighted majority vote learning (Germain et al., 2016).

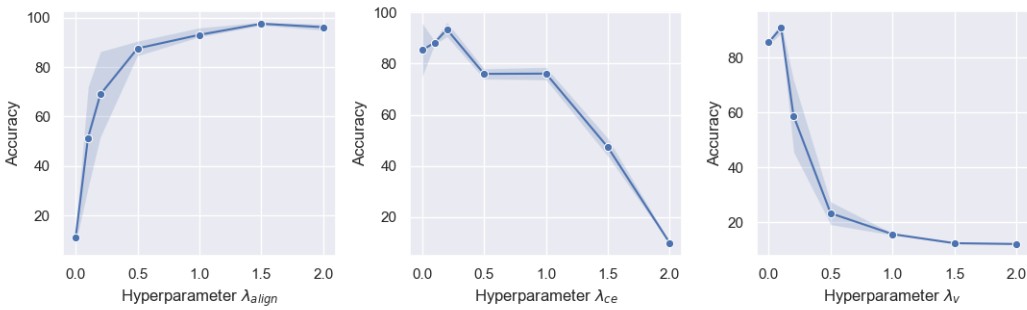

Figure 2: Hyperparameter analysis on USPS→MNIST task

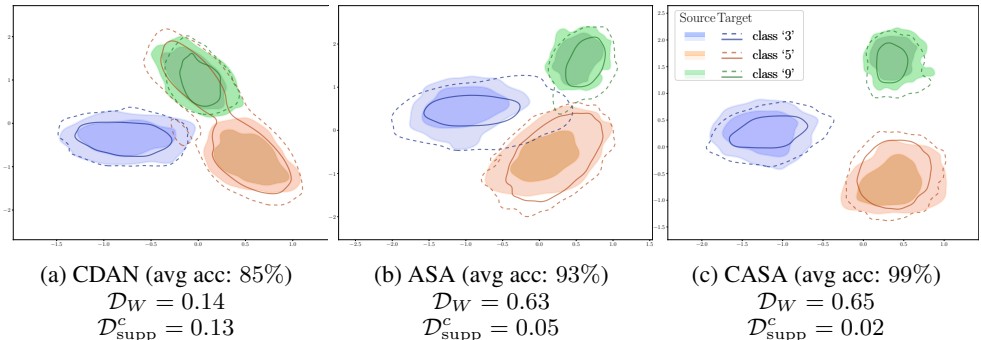

(a) CDAN (avg acc: 85%)
$\mathcal{D}_W = 0.14$
$\mathcal{D}_{\text{supp}}^c = 0.13$

(b) ASA (avg acc: 93%)
$\mathcal{D}_W = 0.63$
$\mathcal{D}_{\text{supp}}^c = 0.05$

(c) CASA (avg acc: 99%)
$\mathcal{D}_W = 0.65$
$\mathcal{D}_{\text{supp}}^c = 0.02$

Figure 3: Visualization of support of feature representations for 3 classes in the USPS → MNIST task. Similar to Tong et al. (2022), each plot illustrates the 2 level sets of kernel density estimates for both the source and target features. The average accuracy, Wasserstein distance $\mathcal{D}_W$ and $\mathcal{D}_{\text{supp}}^c$ are also provided.

Numerous methods have been proposed to align the distribution of feature representation between source and target domains, using Wasserstein distance (Courty et al., 2017; Shen et al., 2018; Lee & Raginsky, 2018), maximum mean discrepancy (Long et al., 2014; 2015; 2016), Jensen-Shannon divergence (Ganin & Lempitsky, 2015; Tzeng et al., 2017), or first and second moment of the concerned distribution (Sun & Saenko, 2016; Peng et al., 2019),.

However, recent works have pointed out the limits of enforcing invariant feature representation distribution, particularly when the marginal label distribution differs significantly between domains (Johansson et al., 2019; Zhao et al., 2019; Wu et al., 2019; Tachet des Combes et al., 2020). Based on these theoretical results, different methods have been proposed to tackle UDA under label shift, often by minimizing $\beta$-relaxed Wasserstein distance (Tong et al., 2022; Wu et al., 2019), or estimating the importance weight of label distribution between source and target domains (Lipton et al., 2018; Tachet des Combes et al., 2020; Azizzadenesheli et al., 2019).

## 5  CONCLUSION

In this paper, we propose a novel CASA framework to handle the UDA problem under label distribution shift. The key idea of our work is to learn a more discriminative and useful representation for the classification task by aligning the supports of the conditional distributions between the source and target domains. We next provide a novel theoretical error bound on the target domain and then introduce a complete training process for our proposed CASA . Our experimental results consistently show that our CASA framework outperforms other relevant UDA baselines on several benchmark tasks. We plan to employ and extend our proposed method to more challenging problem settings, such as domain generalization and universal domain adaptation.

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

# A PROOFS OF THE THEORETICAL RESULTS

## A.1 CSSD AS A SUPPORT DIVERGENCE

*Proof.* First, we aim to demonstrate that $\mathcal{D}_{\text{supp}}^c(P_{Z|Y}^S, P_{Z|Y}^T) \geq 0$ for all $P_{Z|Y}^S$ and $P_{Z|Y}^T$. To establish this, consider any $y \in \mathcal{Y}$:

$$\mathbb{E}_{z \sim P_{Z|Y=y}^S}[d(z, \text{supp}\, P_{Z|Y=y}^T)] = \mathbb{E}_{z \sim P_{Z|Y=y}^S}\left[\inf_{z' \in \text{supp}\, P_{Z|Y=y}^T} d(z, z')\right] \geq 0.$$

This is a consequence of $d(\cdot, \cdot)$ is a distance metric, ensuring $d(z, z') \geq 0$. The same reasoning applies to the second term in the definition of $\mathcal{D}_{\text{supp}}^c(P_{Z|Y}^S, P_{Z|Y}^T)$.

Second, we show that $\mathcal{D}_{\text{supp}}^c(P_{Z|Y}^S, P_{Z|Y}^T) = 0$ if and only if $\text{supp}\, P_{Z|Y=y}^S = \text{supp}\, P_{Z|Y=y}^T$ for any $y \in \mathcal{Y}$. In other words, since $P^S(Y = y) > 0$ and $P^T(Y = y) > 0$, $\mathcal{D}_{\text{supp}}^c(P_{Z|Y}^S, P_{Z|Y}^T) = 0$ if and only if both

$$\mathbb{E}_{z \sim P_{Z|Y=y}^S}[d(z, \text{supp}\, P_{Z|Y=y}^T)] = 0$$

$$\mathbb{E}_{z \sim P_{Z|Y=y}^T}[d(z, \text{supp}\, P_{Z|Y=y}^S)] = 0.$$

The first condition implies that, for any $z \in \text{supp}\, P_{Z|Y=y}^S$, the probability of $d(z, \text{supp}(P_{Z|Y=y}^T)) > 0$ is 0. Consequently, $d(z, \text{supp}\, P_{Z|Y=y}^T) = 0$ for all $z \in \text{supp}\, P_{Z|Y=y}^S$, leading to $\text{supp}\, P_{Z|Y=y}^S \subseteq \text{supp}\, P_{Z|Y=y}^T$. Analogously, the second condition yields $\text{supp}\, P_{Z|Y=y}^T \subseteq \text{supp}\, P_{Z|Y=y}^S$. Combining these, for any $y$, we conclude that $\text{supp}\, P_{Z|Y=y}^T = \text{supp}\, P_{Z|Y=y}^S$. $\qquad\square$

Note that the definition of support divergence is closely related to Chamfer divergence (Fan et al., 2017; Nguyen et al., 2021), which has been shown to not be a valid metric. Figure 1c is best suited to illustrate this proposition as the class-wise supports of two distributions are aligned.

## A.2 LEMMA 1

*Proof.* By the law of total expectation, we can write

$$\text{IMD}_{\mathbb{F}_\epsilon}(P_Z^T, P_Z^S) = \sup_{f \in \mathbb{F}_\epsilon} \mathbb{E}_{P_Y^T} \mathbb{E}_{P_{Z|Y}^T}[f] - \mathbb{E}_{P_Y^S} \mathbb{E}_{P_{Z|Y}^T}[f] = \sup_{f \in \mathbb{F}_\epsilon} \sum_{k=1}^K q_k \mathbb{E}_{P_{Z|Y=k}^T}[f] - p_k \mathbb{E}_{P_{Z|Y=k}^S}[f].$$

Next, we bound the function $f$ using the assumption that $f$ is 1-Lipschitz. That is, for any $z \in \mathcal{Z}$ and $z' \in \text{supp}\, P_{Z|Y=k}^S$, we have

$$f(z) \leq f(z') + d(z, z') \leq \delta_k + d(z, z').$$

The infimum of $d(z, z')$ w.r.t $z' \in \text{supp}\, P_{Z|Y=k}^S$ in the right-hand side will result in $d(z, \text{supp}\, P_{Z|Y=k}^S)$. Therefore, we have

$$f(z) \leq \delta_k + d(z, \text{supp}\, P_{Z|Y=k}^S)$$

Now the class-conditioned expectation of $f$ is bounded by

$$\mathbb{E}_{P_{Z|Y=k}^T}[f] \leq \delta_k + \mathbb{E}_{P_{Z|Y=k}^T}[d(z, \text{supp}\, P_{Z|Y=k}^S)].$$

Together with the definition of $\mathbb{F}_\epsilon$, we can arrive with the first result

$$\text{IMD}_{\mathbb{F}_\epsilon}(P_Z^T, P_Z^S) \leq \sum_{k=1}^K q_k \mathbb{E}_{P_{Z|Y=k}^T}[d(z, \text{supp}\, P_{Z|Y=k}^S)] + q_k \delta_k + p_k \epsilon_k.$$

The second result can be obtained by deriving a similar bound for $\mathbb{E}_{P_{Z|Y=k}^S}[f]$ as

$$\mathbb{E}_{P_{Z|Y=k}^S}[f] \leq \gamma_k + \mathbb{E}_{P_{Z|Y=k}^S}[d(z, \text{supp}\, P_{Z|Y=k}^T)].$$

$\qquad\square$

### A.3 ADDITIONAL ANALYSIS ON $\mathbb{F}_0$

In this section, we demonstrate a special case where given $f \in \mathbb{F}_0$, our bound in Eq equation 3 becomes independent of $\delta_k$. This independence arises due to our significantly relaxed assumption $f \in \mathbb{F}_\epsilon$ and is not directly linked to our proposed CSSD. While the precise interpretation of $\delta_k$ might not immediately clear, the result indicates the trade-off between constraining $\epsilon = 0$ and allowing for $\epsilon > 0$.

Recall that in our proof for Lemma 1, where we can express $\mathrm{IMD}_{\mathbb{F}_0}(P_Z^T, P_Z^S)$ as follows:

$$\mathrm{IMD}_{\mathbb{F}_0}(P_Z^T, P_Z^S) = \sup_{f \in \mathbb{F}_0} \mathbb{E}_{P_Y^T} \mathbb{E}_{P_{Z|Y}^T}[f] - \mathbb{E}_{P_Y^S} \mathbb{E}_{P_{Z|Y}^T}[f] = \sup_{f \in \mathbb{F}_0} \sum_{k=1}^{K} q_k \mathbb{E}_{P_{Z|Y=k}^T}[f] - p_k \mathbb{E}_{P_{Z|Y=k}^S}[f].$$

In the context of $f \in \mathbb{F}_0$, it implies that, for any $z \in \mathrm{supp}\, P_Z^S$, $f(z) = 0$. This also holds for any $z \in \mathrm{supp}\, P_{Z|Y=k}^S \subset \mathrm{supp}\, P_Z^S$, $f(z) = 0$. Using the Lipschitz property, we have, for any $z \in \mathcal{Z}$, $z' \in \mathrm{supp}\, P_{Z|Y=k}^S$,

$$f(z) \leq \underbrace{f(z')}_{=0, \text{no } \delta_k \text{ arises}} + d(z, z') \leq d(z, z').$$

This inequality means $f(z) \leq \mathbb{E}[d(z, \mathrm{supp}\, P_{Z|Y=k}^S)]$ for any $k$. Consequently, we can derive the following bound:

$$\mathrm{IMD}_{\mathbb{F}_0}(P_Z^T, P_Z^S) \leq \sum_{k=1}^{K} q_k \mathbb{E}_{P_{Z|Y=k}^T}[d(z, \mathrm{supp}\, P_{Z|Y=k}^S)].$$

This result aligns precisely with the first term in our CSSD and $\delta_k$ does not appear.

### A.4 PROPOSITION 2

*Proof.* We have $\mathcal{D}_{supp}^c(P_{Z|Y}^S, P_{Z|Y}^T) = 0$ is equivalent to

$$P^S(Z = z | Y = y) > 0 \text{ iff } P^T(Z = z | Y = y) > 0$$

Since $P^S(Y = y) > 0$, and $P^T(Y = y) > 0$, the condition above is equivalent to

$$P^S(Z = z, Y = y) > 0 \text{ iff } P^T(Z = z, Y = y) > 0,$$

which means that

$$\mathcal{D}_{supp}(P_{Z,Y}^S, P_{Z,Y}^T) = 0.$$

$\square$

## B ADDITIONAL COMPARISON TO OTHER GENERALIZED TARGET SHIFT METHODS

The methods proposed in Gong et al. (2016) and Tachet des Combes et al. (2020) both estimate the shifted target label distribution and enforce the conditional domain invariance. However, they rely on several assumptions that may not be practical, e.g., clustering of source and target features, invariant conditional feature distribution between source and target domains, or linear independence of conditional target feature distribution. Similarly, Rakotomamonjy et al. (2022) assumes that there exists a linear transformation between class-conditional distributions in the source and target domains, proposing the use of kernel embedding of conditional distributions to align these distributions. In contrast, our proposed framework does not impose such strict assumptions as those in these prior works and avoids aligning the class-conditional feature distributions.

While the theoretical bound in Tachet des Combes et al. (2020) does not introduce the additional term of $\sum_{k=1}^{K} q_k \delta_k + p_k \gamma_k$ in Theorem 1, our theoretical result does not rely on the strict assumption of

GLS Tachet des Combes et al. (2020), which can be challenging to enforce. Hence, our proposed CASA provides an orthogonal view on the problem of generalized target shift, without imposing stringent assumptions on data distribution shift between source and target domains.

Similar to previously described methods Tachet des Combes et al. (2020); Gong et al. (2016), Rakotomamonjy et al. (2022) proposed learning a feature representation in which both marginals and class-conditional distributions are domain-invariant. The authors also proposed estimating the target label distribution, similar to Gong et al. (2016), in order to align class-conditional feature distribution and thus reduce the target error. Hence, the performance of the algorithm in Rakotomamonjy et al. (2022) relies heavily on accurate estimation of $P_Y^T$, which might be challenging under severe label shift. More importantly, the target error upper bound in Rakotomamonjy et al. (2022) contains the term $sup_{k,z}(w(z)S_k(z))$ that increases together with the severity of label distribution shift, which might degrade the proposed method's performances under severe label shift. In contrast, our bound in Theorem 1 does not have this issue, which may help explain the superior empirical performance of CASA over MARS Rakotomamonjy et al. (2022) under severe label shift that is demonstrated in our global response section.

In Kirchmeyer et al. (2022), the authors proposed learning an optimal transport map between the source and target distribution, as an alternative to the popular approach of enforcing domain invariance. Unlike Kirchmeyer et al. (2022), our method does not require additional assumptions on the source and target feature distribution, including the source domain cluster assumption, and the conditional matching assumption between the source and target domain. While the target risk error bound in Kirchmeyer et al. (2022) contains the Wasserstein-1 divergences between 2 pairs of distribution, one of which is computationally intractable due to the absence of target domain labels, our proposed bound contains only the support divergence between conditional source and target feature distribution. Because the support divergence has been shown to be considerably smaller than other conventional distribution divergences, e.g. Wasserstein-1 divergence, the proposed error bound can be tighter than that of Kirchmeyer et al. (2022). Moreover, the last term in the bound of Kirchmeyer et al. (2022) is inversely proportional to the minimum proportion of a particular class in the target domain, making the performance of OSTAR degrade considerably on severe label shift Kirchmeyer et al. (2022). On the contrary, our bound does not suffer from such issue on severe label shift. However, the trade-off for the absence of additional assumptions like those in Kirchmeyer et al. (2022) is that our bound introduces an additional term of $\sum_{k=1}^{K} q_k \delta_k + p_k \gamma_k$, which intuitively is the sum of a worst-case per-class error on both source and target domain. As we mentioned in Remark 3, we assume this term and the ideal joint risk term to be small, similar to existing domain adversarial methods Ben-David et al. (2006); Ganin et al. (2016), and minimize the first and second terms in our bound.

## C  ADDITIONAL EXPERIMENT RESULTS

We further conduct experiments on the DomainNet dataset, following the same experiment setting in the main paper, and report the results in Table 5. Overall, while CASA provides lower results under $\alpha \in \{None, 10.0\}$ than FixMatch(RS+RW) and SDAT, CASA consistently achieves the highest accuracy scores under more severe label shift setting.

## D  DATASET DESCRIPTION

- **USPS → MNIST** is a digits benchmark for adaptation between two grayscale handwritten digit datasets: USPS (Hull, 1994) and MNIST (LeCun et al., 1998). In this task, data from the USPS dataset is considered the source domain, while the MNIST dataset is considered the target domain.

- **STL → CIFAR**. This task considers the adaptation between two colored image classification datasets: STL (Coates & Ng, 2012) and CIFAR-10 (Krizhevsky et al., 2009). Both datasets consist of 10 classes of labels. Yet, they only share 9 common classes. Thus, we adapt the 9-class classification problem proposed by Shu et al. (2018) and select subsets of samples from the 9 common classes.

- **VisDA-2017** is a synthetic to real images adaptation benchmark of the VisDA-2017 challenge (Peng et al., 2017). The training domain consists of CAD-rendered 3D models of 12 classes of objects from different angles and under different lighting conditions. We use the

Table 5: Per-class accuracy on DomainNet

| Algorithm | $\alpha =$ None | $\alpha = 10$ | $\alpha = 3.0$ | $\alpha = 1.0$ | $\alpha = 0.5$ | Average |
|---|---|---|---|---|---|---|
| No DA | 40.6 | 40.6 | 40.6 | 40.5 | 40.5 | 40.6 |
| DANN* | 44.2 | 43.7 | 43.0 | 43.0 | 40.8 | 43.0 |
| CDAN* | 44.4 | 43.8 | 43.2 | 43.1 | 41.6 | 43.2 |
| VADA | 44.0 | 43.6 | 43.3 | 42.6 | 42.3 | 43.2 |
| FixMatch* | **45.1** | **44.5** | 43.3 | 42.9 | 42.3 | 43.6 |
| SDAT | 44.6 | 44.2 | 43.2 | 42.0 | 41.2 | 43.0 |
| MIC | 44.8 | 44.3 | 43.3 | 42.2 | 41.1 | 43.1 |
| IWDAN | 43.8 | 43.2 | 41.5 | 39.2 | 38.4 | 41.3 |
| IWCDAN | 44.0 | 44.1 | 43.2 | 42.8 | 41.7 | 43.2 |
| sDANN | 43.6 | 43.3 | 43.5 | 43.1 | 42.6 | 43.3 |
| ASA | 42.9 | 41.8 | 41.3 | 39.6 | 39.3 | 41.0 |
| PCT | 44.6 | 43.5 | 43.3 | 42.1 | 40.4 | 42.8 |
| SENTRY | 43.4 | 43.4 | 43.1 | 42.5 | 42.1 | 42.9 |
| CASA | 44.5 | 44.1 | **43.6** | **43.4** | **42.9** | **43.7** |

validation data of the challenge, which consists of objects of the same 12 classes cropped from images of the MS COCO dataset (Lin et al., 2014), as the target domain.

- **DomainNet** dataset contains about 0.6 million images in total with 345 classes (Peng et al., 2019). We consider 3 domains from this dataset: real, painting and sketch, use the real domain as the source and the other 2 domain as targets.

# E  IMPLEMENTATION DETAILS

**USPS → MNIST**. Following Tachet des Combes et al. (2020), we employ a LeNet-variant (LeCun et al., 1998) with a 500-d output layer as the backbone architecture for the feature extractor. For the discriminator, we implement a 3-layer MLP with 512 hidden units and leaky-ReLU activation.

We train all classifiers, along with their feature extractors and discriminators, using 65000 SGD steps with learning rate 0.02, momentum 0.9, weight decay $5 \times 10^{-4}$, and batch size 64. The discriminator is updated once for every update of the feature extractor and the classifier. After the first 30000 steps, we apply linear annealing to the learning rate for the next 30000 steps until it reaches the final value of $2 \times 10^{-5}$.

For the loss of the feature extractor, the alignment weight $\lambda_{align}$ is scheduled to linearly increase from 0 to 1.0 in the first 10000 steps for all alignment methods, and $\lambda_{vat}$ equals 1.0 for the source, and 0.1 for the target domains.

**STL → CIFAR.** We follow Tong et al. (2022) in using the same deep CNN architecture as the backbone for the feature extractor. The 192-d feature vector is then fed to a single-layer linear classifier. The discriminator is a 3-layer MLP with 512 hidden units and leaky-ReLU activation.

We train all classifiers, along with their feature extractors and discriminators, using 40000 ADAM (Kingma & Ba, 2015) steps with learning rate 0.001, $\beta_1 = 0.5$, $\beta_2 = 0.999$, no weight decay, and batch size 64. The discriminator is updated once for every update of the feature extractor and the classifier.

For the loss of the feature extractor, the weight of the alignment term is set to a constant $\lambda_{align} = 0.1$ for all alignment methods. The weight of the auxiliary conditional entropy term is $\lambda_{ce} = 0.1$ for all domain adaptation methods, and $\lambda_{vat}$ equals 1.0 for the source, and 0.1 for the target domains.

**VisDA-2017.** We use a modified ResNet-50 (He et al., 2016) with a 256-d final bottleneck layer as the backbone of our feature extractor. All layers of the backbone, except for the final one, use pretrained weights from `torchvision` model hub. The classifier is a single linear layer. Similar to other tasks, the discriminator is a 3-layer MLP with 1024 hidden units and leaky-ReLU activation.

We train all classifiers, feature extractors, and discriminators using 25000 SGD steps with momentum 0.9, weight decay 0.01, and batch size 64. We use a learning rate of 0.001 for feature extractors. For the classifiers, the learning rate is 0.01. For the discriminator, the learning rate is 0.005. We apply linear annealing to the learning rate of feature extractors and classifiers such that their learning rates are decreased by a factor of 0.05 by the end of training.

The alignment weight $\lambda_{align}$ is scheduled to linearly increase from 0 to 0.1 in the first 5000 steps for all alignment methods. The weight of the auxiliary conditional entropy term is set to a constant $\lambda_{ce} = 0.05$, and $\lambda_{vat}$ equals 0 for the source, and 0.1 for the target domains.

**DomainNet.** We use the same backbone and network architecture as those of VisDA-2017 experiments. We train all classifiers, feature extractors, and discriminators using 20000 SGD steps with momentum 0.9, weight decay 0.0001, and batch size 64. We use a learning rate of 0.01 for feature extractors. For the classifiers, the learning rate is 0.1. For the discriminator, the learning rate is 0.01. We use the same learning rate scheduler as that of Garg et al. (2023). The values for $\lambda_{align}$, $\lambda_{ce}$ and $\lambda_v$ are 1.0, 0.02 and 0.1, respectively.

