# OpenReview forum: "Conditional Support Alignment for Domain Adaptation with Label Shift"
_ICLR.cc/2024/Conference — Submitted to ICLR 2024_

### Official Review · Reviewer_o6SV · 2023-10-21

**Soundness:** 3 good
**Presentation:** 3 good
**Contribution:** 2 fair
**Rating:** 6
**Confidence:** 3

**Summary:**

This paper proposes Conditional Adversarial Support Alignment (CASA) whose aim is to minimize the Conditional Symmetric Support Divergence (CSSD) between the source’s and target domain’s feature representation distributions, aiming at a more discriminative representation for the classification task. Theoretical analyses are also provide in this work.

**Strengths:**

1. The proposed CASA addresses the drawback of Adversarial Support Alignment (ASA) by considering discriminative features to align the supports of two distributions, thus mitigating the risk of conditional distribution misalignment caused by indiscriminate reduction of marginal support divergence.
2. Theoretical target error bound are provided in this work.
3. Extensive experiments are conducted to demonstrate the effectiveness of the proposed method.

**Weaknesses:**

1. The major difference between this work and ASA is the conditional alignment, specifically CSSD and SSD. However, the label in the target domain is unknown, and the authors utilize the entropy conditioning technique described in [1] to address this issue. As far as I know, the method in [1] is not specifically designed for generating pseudo-labels. Could the authors please explain how they adapt this method to mitigate the error accumulation problem associated with using pseudo-labels? A detailed explanation from the authors would be appreciated.
2. More SOTA methods are suggested to discuss and compare, such as SHOT [2], BIWAA [3], CoVi [4], etc.


[1] Long, M., Cao, Z., Wang, J., & Jordan, M. I. (2018). Conditional adversarial domain adaptation. Advances in neural information processing systems, 31.

[2] Liang, J., Hu, D., & Feng, J. (2020, November). Do we really need to access the source data? source hypothesis transfer for unsupervised domain adaptation. In International conference on machine learning (pp. 6028-6039). PMLR.

[3] Westfechtel, T., Yeh, H. W., Meng, Q., Mukuta, Y., & Harada, T. (2023). Backprop Induced Feature Weighting for Adversarial Domain Adaptation with Iterative Label Distribution Alignment. In Proceedings of the IEEE/CVF Winter Conference on Applications of Computer Vision (pp. 392-401).

[4] Na, J., Han, D., Chang, H. J., & Hwang, W. (2022, October). Contrastive vicinal space for unsupervised domain adaptation. In European Conference on Computer Vision (pp. 92-110). Cham: Springer Nature Switzerland.

**Questions:**

see Weakness

---

> ### Author Response · Authors · 2023-11-18
>
> We greatly appreciate the reviewer's positive evaluation and constructive comments and suggestions. In the following, we provide the response to your main comments.
>
> ### Explanation for mitigating the accumulative error by using pseudo-labels from CDAN
>
> Theorem 1 motivates us to minimize the conditional support divergence between the source and target domain, in order to minimize the target error risk.
> However, as mentioned in our main paper, due to the absence of labels in the target domain, the proposed CASA method resorts to using pseudo-labels on the target domain.
> Based on several works that have pointed out the error accumulation issue in naively adopting pseudo-labels under data distribution shift [], we utilize the entropy conditioning technique proposed in [1] to mitigate this issue.
> Intuitively, we reweight the importance of each sample in Eq. 9 and Eq. 10 by the quality of their pseudo-labels, which is measured by pseudo-labels' entropy $w_{i} = 1 + e^{-H(g(x_{i}))}$ [1].
> Since the prediction's entropy is closely related to model's error under domain shift [2], this reweighting scheme allows the model to prioritize easily transferable samples, and lessen the issue of error accumulation under domain shift.
> On the other hand, more advanced techniques on generating high-quality pseudo-labels [3][4][5] can be utilized in our model, and potentially help achieve even higher results.
> Investigating closely the effects of different pseudo-labels methods is outside the scope of our paper, and we leave this exploration to future works.
>
> $$
> L_{d}(\phi) = - \frac{1}{n_S}\sum_{i=1}^{n_{S}} w_{i}^S\ln \left[G(x_i^S)\right]-\frac{1}{n_T}\sum_{i=1}^{n_{T}} w_{i}^T \ln \left[1-G(x_i^T)\right]
> $$
>
> $$
> L_{align}(f) = \frac{1}{n_S}\sum_{i=1}^{n_{S}} w_{i}^S d\left(G(x_i^S),\{G(x_j^T)\}_{j=1}^{n_T}\right)
> $$
>
> $$
> +\frac{1}{n_T}\sum_{i=1}^{n_{T}}  w_{i}^T d\left(G(x_i^T),\{G(x_j^S)\}_{j=1}^{n_S}\right)
> $$
>
>
>
> ### Comparison with more SOTA methods: SHOT, BIWAA, and CoVi
> Please refer to the additional experimental results mentioned in our global response. As expected, those new results show that our proposed CASA consistently outperforms others by a large margin.
>
>
> [1] Long et al. Conditional adversarial domain adaptation. Advances in neural information processing systems, 31.
>
> [2] Dequan Wang et al. “Tent: Fully test-time adaptation by entropy minimization”. In: arXiv preprint arXiv:2006.10726
>
> [3] Jian Liang, Dapeng Hu, and Jiashi Feng. “Do we really need to ac- cess the source data? source hypothesis transfer for unsupervised domain adaptation”. In: International Conference on Machine Learning. PMLR. 2020, pp. 6028–6039.
>
> [4] Hong Liu, Jianmin Wang, and Mingsheng Long. “Cycle self-training for domain adaptation”. In: Advances in Neural Information Processing Systems 34 (2021), pp. 22968–22981.
>
> [5] Qiming Zhang et al. “Category anchor-guided unsupervised domain adaptation for semantic segmentation”. In: Advances in Neural Information Processing Systems 32 (2019)

---

> ### Author Response · Authors · 2023-11-22
>
> Dear reviewer,
>
> Thank you once again for taking the time to read and review our submission. We would be happy to address any remaining questions before the discussion period ends today.
>
> Best regards,
>
> Authors.

---

> > ### Comment · Reviewer_o6SV · 2023-11-22
> > **Thanks for authors' responses.**
> >
> > Thanks for authors' responses. I keep my score.

---

> > > ### Author Response · Authors · 2023-11-23
> > > **Thanks for your post-rebuttal response**
> > >
> > > Thank you very much for your great support. We appreciate your time, valuable feedback, and constructive suggestions for improvement in our submission.

---

### Official Review · Reviewer_UE75 · 2023-10-25

**Soundness:** 2 fair
**Presentation:** 2 fair
**Contribution:** 2 fair
**Rating:** 6
**Confidence:** 4

**Summary:**

This paper studies the distribution shift problem for the machine learning model. Specifically, the authors consider the label shift scenario and analyze the limitations in current label shift research, i.e., the strict identical assumption on the conditional distribution $P_{X|Y}$. To address this problem, a novel metric is developed based on the symmetric support divergence (SSD). Mathematically, the proposed metric can be taken as the sliced SSD on each conditional distribution $P_{X|Y=y}$. A new generalization upper bound and some theoretical properties of the proposed metric are provided, which ensure the metric-based model can reduce the generalization error and show the relation between marginal SSD and conditional SSD. Experiments are conducted to show the superiority of the proposed method.

**Strengths:**

+ A conditional variant of SSD and corresponding theoretical analysis are provided.
+ A discrepancy optimization model is proposed to address the domain adaptation with label shift.
+ Superior experiment results are achieved.

**Weaknesses:**

- The basic problem in this paper is indeed equivalent to the generalized target/label shift, where label distribution and conditional distribution change simultaneously. However, many important and closely related references are not introduced and discussed.
- Consider the existing results for generalized target/label shift, the generalization error analysis provided in Thm. 1 seems to be less compact and not informative.
- Important theoretical results for the main merits, i.e., conditional variant of SSD, are missing, which makes the proposed method less technically sound.
- The organization and clarity should be improved. Some justification and intuition for the math definition or theoretical results are insufficient.
- The experiment comparison is insufficient, where some related works are omitted.

**Questions:**

1. The essential setting and problem that are considered in this submission is indeed similar to the well-known generalized target/label shift [a-f], which are not properly introduced and discussed. Besides, the label shift problem is also extensively studied and has shown promising theoretical results in many studies. From both the generalized target/label shift view and label shift view, this paper does not provide sufficient discussion with these existing methodological and theoretical results. Thus, it is hard to evaluate this paper's contributions, making this work less persuasive.

2. In the generalized target/label shift literature [e, f], generalization bounds and theoretical analysis are also provided. Compared with these results that compactly decompose the shift on the joint distribution as the terms determined by label discrepancy and conditional discrepancy, this paper induces additional constants, i.e., joint error on both domains and the non-negative constants $\delta, \gamma$ induced by IMD.

3. Considering the existing results, the main contribution in this paper is the new conditional discrepancy metric. However, it seems that it cannot be rigorously considered as the class-wise IMD. Specifically, note for the IMD in Def. 3, the weights of the two expected divergence terms are 1; however, in the conditional variant in Def. 4, the divergence terms are weighted by the label probability masses $P(Y=y)$. In such a definition, it naturally raises an crucial question, i.e., is the conditional SSD in Def.4 defines a metric on conditional distribution? This theoretical property is the foundation for the proposed method and should be treated rigorously.

4. The justifications of the derived theoretical results should be improved. Though Thm. 1 ensures that the generalized label shift correction is sufficient to mitigate the label discrepancy and conditional discrepancy, the constants induced in upper-bound seem to be intractable.

5. The discussion in Remark 3 is insufficient and seems to be improper. The advantages of existing results [e] are not properly stated, i.e., literature [e] does not induce additional constant that cannot be controlled by the learning model. Besides, the related works [d,f] are not discussed and compared.

6. Since there are many related works in correcting label shift and conditional shift simultaneously [a-f], they should also be carefully compared in experiment validation.

[a] Zhang, Kun, et al. "Domain adaptation under target and conditional shift." International conference on machine learning. PMLR, 2013.

[b] Gong, Mingming, et al. "Domain adaptation with conditional transferable components." International conference on machine learning. PMLR, 2016.

[c] Ren, Chuan-Xian, Xiao-Lin Xu, and Hong Yan. "Generalized conditional domain adaptation: A causal perspective with low-rank translators." IEEE transactions on cybernetics 50.2 (2018): 821-834.

[d] Rakotomamonjy, Alain, et al. "Optimal transport for conditional domain matching and label shift." Machine Learning (2022): 1-20.

[e] Tachet des Combes, Remi, et al. "Domain adaptation with conditional distribution matching and generalized label shift." Advances in Neural Information Processing Systems 33 (2020): 19276-19289.

[f] Kirchmeyer, Matthieu, et al. "Mapping conditional distributions for domain adaptation under generalized target shift." International Conference on Learning Representations. 2022.

---

> ### Author Response · Authors · 2023-11-18
>
> We greatly appreciate the reviewer’s valuable comments and constructive suggestions. We address the main concerns as follows.
>
> ### Comparison to other relevant theoretical works
> We acknowledge the reviewer's concern over the lack of comparison with existing theoretical results on the generalized label shift problem. Our paper previously only focused on comparing the derived theoretical results with the existing literature on adversarial domain adaptation [3] [4], and adversarial support alignment [1] in section 2.3. To provide a more comprehensive treatment of existing approaches to  domain adaptation with generalized target shift, we discuss the result of Theorem 1 and other relevant theoretical results in Section B of the updated Appendix. Thus, we refer the reviewer to this section in the revised paper for more detailed discussion.
>
>
> ### Tractability of the induced constants in Theorem 1
> We agree with the reviewer that those constants might be computationally intractable. Nevertheless, in the theoretical development of our method, we assume the existence of those constants to obtain the exact theoretical upper bound of the target error. In fact, those induced constants remain unchanged in the training process of our CASA, whose main goal is to reduce the divergence term $D_{supp}^c(P^S_{Z|Y},P^T_{Z|Y})$. In particular, on the additional induced quantity $\delta$, the appearance of additional term is due to the mild assumption ${f \in F}_{\epsilon}$ where $\epsilon$ be greater than 0. In fact, if we impose a more stringent assumption by setting $\epsilon$ to be precisely $0$ (indicating models making perfect predictions on labeled source data), these quantities will vanish. Please refer our updated Appendix for further details
>
> ### Is conditional SSD in Def.4 a metric on conditional distribution?
>
> The conditional symmetric support divergence (CSSD) $D_{supp}^c(P^S_{Z|Y},P^T_{Z|Y})$ is not a proper metric on conditional distribution; instead it serves a support divergence on conditional distribution. This is similar to [1] where their proposed symmetric support divergence is similarly categorized as a support divergence. The definition of support divergence is closely related to Chamfer divergence [1][2], which has been shown to not be a valid metric, since it does not satisfy the triangle inequality. In response to reviewer's suggestion, we have added Proposition 1 and the additional analysis to this on in the updated Appendix. Our approach distinguishes itself by targeting the alignment of supports in class-wise conditional distributions rather than those in margin distributions. Intuitively, two distributions $P_{Z|Y}, Q_{Z|Y}$ with the same label space can have $D^c_{supp}(P_{Z|Y}, Q_{Z|Y}) = 0$, if and only if their conditional supports are equal, i.e. $supp(P_{Z|Y=y}) = supp(Q_{Z|Y=y}) \forall y$.
>
> *We sincerely hope that you can reconsider the review score. Please let us know if there are further things you would like us to address.*
>
> *Best regards,*
>
> *Authors*
>
> [1] Shangyuan Tong et al. “Adversarial Support Alignment”. In: arXiv preprint arXiv:2203.08908 (2022)
>
> [2] Haoqiang Fan et al. A point set generation network for 3d object reconstruction from a single image. In Proceedings of the IEEE conference on computer vision and pattern recognition, pp. 605–613, 2017
>
> [3] Shai Ben-David et al. “Analysis of representations for domain adaptation”. In: Advances in neural information processing systems 19 (2006)
>
> [4] David Acuna et al. “f-domain adversarial learning: Theory and algorithms”. In: International Conference on Machine Learning. PMLR. 2021, pp. 66–75.

---

> > ### Comment · Reviewer_UE75 · 2023-11-21
> > **Thanks for the detailed responses.**
> >
> > I want to thank the authors for their comprehensive responses, where the questions on theoretical results are addressed and the existing results are properly compared. After checking the revision, I think the technical novelty of the paper is clear, which mainly focuses on the theoretical advances in *the support divergence* and shows merits compared with existing works. Therefore, I will improve the score to 6 for acceptance.

---

> > > ### Author Response · Authors · 2023-11-21
> > > **Thank you so much for upgrading the paper score**
> > >
> > > Dear reviewer,
> > >
> > > We are grateful for your post-rebuttal response, especially for your reconsideration. We are also glad that most of your initial concerns about our paper have been addressed.
> > >
> > > Best regards,
> > >
> > > Authors.

---

### Official Review · Reviewer_XqJP · 2023-11-01

**Soundness:** 3 good
**Presentation:** 3 good
**Contribution:** 2 fair
**Rating:** 5
**Confidence:** 4

**Summary:**

This paper proposed conditional adversarial support alignment (CASA) to minimize the conditional symmetric support divergence between the source’s and target domain’s feature representation distributions. Generally, the paper is well-written and easy to follow. They evaluate the model on several benchmarks from various types of results. However, the model's novelty is incremental.

**Strengths:**

This paper proposed conditional adversarial support alignment (CASA) to minimize the conditional symmetric support divergence between the source’s and target domain’s feature representation distributions. Generally, the paper is well-written and easy to follow. They evaluate the model on several benchmarks from various types of results.

**Weaknesses:**

The model's novelty is incremental over multiple loss functions. The alignment loss in Eq(10) is more like pair-wise alignment loss, which has been explored before for cross-domain graph alignment. It is hard to verify the novelty.

From the experiments, they show the improvements when \alpha decreases. However, there is no insight why this happens. It is better to discuss the intuition and data used behind. It needs more visualization to demonstrate the improvement.

**Questions:**

The novelty clarification.
The performance analysis.

---

> ### Author Response · Authors · 2023-11-18
>
> We greatly appreciate the reviewer’s valuable comments and constructive suggestions. We address the main concerns as follows.
>
> ### Novelty clarification
>
> We would like to emphasize that our main contribution includes a novel conditional SSD-based domain adaptation bound in section 2.3. The main motivation for our proposed method is to utilize discriminative features during adversarial domain alignment to improve performance under different levels of label distribution shift. Based on the novel target error bound in section 2.3, we then propose the corresponding training scheme that empirically yields superior results on benchmark datasets under the setting of label distribution shift. We provide a detailed comparison of our proposed error bound to other relevant theoretical results on the problem of unsupervised domain adaptation, in Remark 3, and in our response to **Reviewer UE75**. The corresponding loss terms in Eq. 7, 8, and 10 are directly motivated by our proposed bound in Theorem 1. Furthermore, we provide an analysis of these three loss terms in section 3.3 and confirm the merits of optimizing each of these loss terms in our ablation study.
>
> ### Similarity between CSSD and pair-wise graph alignment
>
> We thank the reviewer for this suggestion and would appreciate it if the reviewer could point out which specific work on cross-domain graph alignment he/she is referring to. However, as pointed out in [1], the support alignment loss is closely related to the Chamfer divergence, which also has been used extensively in 3D point cloud modeling [4, 5] and learning document embedding [6]. Different from these works, ours is the first work that proposes using the conditional support divergence within the problem domain of domain adaptation with generalized target shift. Furthermore, we have provided a comprehensive discussion of our novel alignment loss in section 2.3, and the novel optimization scheme of this loss term in section 2.4.3 in our main paper.
>
> ### More performance analysis
>
> We thank the reviewer for the suggestion of clarifying the competitive performance of CASA over severe label shift. However, we would like to point out that, in fact, CASA's performance does not improve as alpha decreases, as **Reviewer XqJP** has claimed. In particular, Tables 1-3 show that more severe levels of label shift also degrade CASA's performance mildly on all of the benchmark datasets. More importantly, Tables 1-3 show that CASA performs competitively with other baselines under alpha=\{None, 10.0\}, and CASA attains the highest results under other smaller values of alphas. There are several reasons for this phenomenon. Since the objective of minimizing CSSD does not significantly degrade the model's performance under large label shifts, as we already discussed in Remark 3 and in our response to **Reviewer UE75**, CASA is more robust to large label distribution shifts. On the other hand, as support divergence is an extremely relaxed form of distribution divergence [1], aligning the supports of source and target features may not perform as well as aligning these distributions directly when there is no or mild label shift. We have visualized the feature distributions and studied the performance of CASA, in comparison to 2 other baselines CDAN and ASA in Figure 3 and Section 3.3. Compared to CDAN, the Wasserstein distance between source and target features of CASA is much larger, at a value of 0.65. This helps explain the superior performance of CASA over CDAN and other similar distribution-alignment methods under severe label shift of small alpha values [2,3].
>
> We really hope that you can reconsider the review score. Please let us know if you would like us to do anything else.
>
> Best regards,
>
> Authors
>
>
>
> [1] Shangyuan Tong et al. “Adversarial Support Alignment”. In: arXiv preprint arXiv:2203.08908 (2022)
>
> [2] Zhao et al. On learning invariant
> representations for domain adaptation. In International Conference on Machine Learning, pp.7523–7532. PMLR, 2019
>
> [3] Johansson et al. Support and invertibility in domain-invariant representations. In The 22nd International Conference on Artificial Intelligence and Statistics, pp. 527–536. PMLR, 2019.
>
> [4] Haoqiang Fan et al. A point set generation network for 3d object reconstruction from a single image. In Proceedings of the IEEE conference on computer vision and pattern recognition, pp. 605–613, 2017
>
> [5] Trung Nguyen et al. “Point-set distances for learning representations of 3d point clouds”. In: Proceedings of the IEEE/CVF International Conference on Computer Vision. 2021, pp. 10478–10487
>
> [6] Matt Kusner et al. From word embeddings to document distances. In International conference on machine learning, pp. 957–966. PMLR, 2015

---

> ### Author Response · Authors · 2023-11-22
>
> Dear reviewer,
>
> Thank you once again for taking the time to read and review our submission. We would be happy to address any remaining questions before the discussion period ends today.
>
> Best regards,
>
> Authors.

---

> > ### Author Response · Authors · 2023-11-23
> >
> > Dear Reviewer XqJP,
> >
> > Thanks a lot for your efforts in reviewing this paper. We have tried our best to address all your concerns and provided clarifications on all confusing concepts.
> >
> > The discussion period is nearing an end (in only a few hours), thus we wonder if you can spend some time going over our newest replies, to see if they successfully answered your questions or not. This is also to give us a decent amount of time to address any of your remaining/additional concerns.
> >
> > Best regards,
> >
> > Authors.

---

### Author Response · Authors · 2023-11-18
**Global response to concerns about additional baselines' experiments**

We thank all reviewers for the valuable feedback. We appreciate that our paper is well-written and easy to follow (**Reviewer XqJP**), novel (**Reviewer UE75**), theoretically sound (**Reviewer UE75, o6SV**), extensive experiments, and achieving convincing empirical results (all reviewers).

Regarding the common requirement of the reviewers for comparisons between our proposed CASA and other related methods, it is worth noting that, in our main paper, we have included a reasonable number of relevant methods that consider generalized label shift, e.g. IWDAN, IWCDAN (NeurIPS’20), sDANN (ICML’19), ASA (ICLR’22), PCT (NeurIPS’21), SENTRY (ICCV’21), and recent SOTA UDA methods, e.g. FixMatch (NeurIPS’20), SDAT (ICML’22), MIC (CVPR’23). Furthermore, compared to the baseline methods mentioned by **Reviewer UE75** and **Reviewer o6SV**, only ours include comprehensive experiment results on an extensive range of label shift levels, on two relatively large-scale UDA datasets including Visda17 and Domainnet. Nevertheless, we acknowledge the concerns of **Reviewer UE75** and **Reviewer o6SV** over the lack of experiment results with more relevant SOTA baselines. Thus, we have conducted additional experiments on SHOT [2], OSTAR [1], MARS [4], BIWAA [3], and DALN [5] on the large-scale image recognition dataset Visda17, with the same experiment setting as that of the main paper. Those new results are reported in Table 1. On the other hand, we could not perform experiments on the methods in [6, 7, 8] mentioned by **Reviewer UE75**, as the official code repositories seemingly are not publicly released, and we could not reproduce these methods given the time constraints of ICLR rebuttal phase. Similarly, we could not reproduce CoVi [9] due to unaddressed issues in the official code repositories, and performed experiments on SOTA method DALN [5] instead.


|Algorithm|$\alpha=None$|$\alpha=10.0$|$\alpha=3.0$|$\alpha=1.0$|$\alpha=0.5$|avg |
|---------|------------|------------|-----------|-----------|-----------|----|
|No DA    |55.6        |56.0        |55.5       |55.2       |55.1       |55.5|
|SHOT     |72.2        |71.5        |68.1       |62.3       |57.2       |66.3|
|OSTAR    |67.7        |66.8        |65.3       |60.8       |56.3       |63.4|
|MARS     |62.4        |61.2        |59.5       |57.7       |55.5       |59.3|
|BIWAA    |73.2        |72.7        |69.9       |62.3       |60.8       |67.8|
|DALN     |73.4        |72.8        |69.7       |62.9       |55.4       |66.8|
|**CASA**     |**74.3**        |**73.4**        |**71.8**       |**66.3**       |**63.2**       |**69.8**|

Overall, the new results consistently show that the CASA achieves the highest average performance across different levels of label shift among all tested methods. More important, under the most severe label shift, i.e., α = 0.5, CASA’s accuracy is approximately 7-8% higher than those of OSTAR [1] and MARS [4] that explicitly tackle the generalized target shift issue. This observation is consistent with the theoretical results in [1] and [4], i.e., Theorem 1 in [4] and Theorem 1 in [1], which show that these derived target error bounds may increase with more severe marginal label shifts. Our theoretical bound does not suffer from this issue, and the empirical superiority of CASA over OSTAR and MARS is consistent with this theoretical result.


[1] Kirchmeyer, Matthieu, et al. "Mapping conditional distributions for domain adaptation under generalized target shift." International Conference on Learning Representations. 2022.

[2] Liang et al. Do we really need to access the source data? source hypothesis transfer for unsupervised domain adaptation. In International conference on machine learning (pp. 6028-6039). PMLR.

[3] Westfechtel et al. Backprop Induced Feature Weighting for Adversarial Domain Adaptation with Iterative Label Distribution Alignment. In Proceedings of the IEEE/CVF Winter Conference on Applications of Computer Vision (pp. 392-401).

[4] Rakotomamonjy, Alain, et al. "Optimal transport for conditional domain matching and label shift." Machine Learning (2022): 1-20.

[5] Lin Chen et al. “Reusing the Task-specific Classifier as a Discriminator: Discriminator-free Adversarial Domain Adaptation”. In: Proceedings of the IEEE/CVF Conference on Computer Vision and Pattern Recognition. 2022, pp. 7181–7190

[6] Zhang, Kun, et al. "Domain adaptation under target and conditional shift." International conference on machine learning. PMLR, 2013.

[7] Gong, Mingming, et al. "Domain adaptation with conditional transferable components." International conference on machine learning. PMLR, 2016.

[8] Ren et al. "Generalized conditional domain adaptation: A causal perspective with low-rank translators." IEEE transactions on cybernetics 50.2 (2018): 821-834.

[9] Na et al. Contrastive vicinal space for unsupervised domain adaptation. In European Conference on Computer Vision (pp. 92-110).

---

### Meta-Review · Area_Chair_vuKk · 2023-12-07

**Metareview:**

The paper addresses the problem of generalized target shift in domain adaptation problems. It introduces
a new framework that seeks at  minimizing  the conditional symmetric support divergence between the source’s and target domain’s feature representation distributions.

Reviewers have been lukewarm about the paper and they have a mix feeling about it. As an expert in domain adaptation,
I tend to agree with the decision that the paper is bordeline on the reject side.  We believe that the paper
can be made stronger by comparing to recent methods as proposed in the rebuttal. However, an in-depth analysis (ablation study)
on why the proposed method works better than SHOT [2], OSTAR [1], MARS [4], BIWAA [3], and DALN [5] (the first paper does
resampling for avoiding label shift, the two others align conditional distributions while estimating the label proportion, ...)
will make the paper far stronger in addition to clarification on the contributions of each loss in the model.

**Justification For Why Not Higher Score:**

lack of in-depth analysis of the method with respect to competitors

**Justification For Why Not Lower Score:**

na

---

### Decision · Program_Chairs · 2024-01-16

Reject